# An Integrated Approach on the Diagnosis of Cerebral Veins and Dural Sinuses Thrombosis (a Narrative Review)

**DOI:** 10.3390/life12050717

**Published:** 2022-05-11

**Authors:** Dragos Catalin Jianu, Silviana Nina Jianu, Traian Flavius Dan, Georgiana Munteanu, Alexandra Copil, Claudiu Dumitru Birdac, Andrei Gheorghe Marius Motoc, Any Docu Axelerad, Ligia Petrica, Sergiu Florin Arnautu, Raphael Sadik, Nicoleta Iacob, Anca Elena Gogu

**Affiliations:** 1Department of Neurosciences-Division of Neurology, Victor Babes University of Medicine and Pharmacy, E. Murgu Sq., no.2, 300041 Timisoara, Romania; jianu.dragos@umft.ro (D.C.J.); georgiana.munteanu@umft.ro (G.M.); anca.gogu@umft.ro (A.E.G.); 2Centre for Cognitive Research in Neuropsychiatric Pathology (NeuroPsy-Cog), Department of Neurosciences, Victor Babes University of Medicine and Pharmacy, 156 L. Rebreanu Ave., 300736 Timisoara, Romania; alexandra.copil@yahoo.com (A.C.); claudiubirdac8@gmail.com (C.D.B.); amotoc@umft.ro (A.G.M.M.); axelerad.docu@365.univ-ovidius.ro (A.D.A.); petrica.ligia@umft.ro (L.P.); arnautu.sergiu@umft.ro (S.F.A.); 3First Department of Neurology, Pius Branzeu Clinical Emergency County Hospital, 156 L. Rebreanu Ave., 300736 Timisoara, Romania; 4Centre for Molecular Research in Nephrology and Vascular Pathology, Department of Internal Medicine II, Victor Babes University of Medicine and Pharmacy, 156 L. Rebreanu Ave., 300736 Timisoara, Romania; 5Department of Ophthalmology, Dr. Victor Popescu Military Emergency Hospital, 7 G. Lazar Ave, 300080 Timisoara, Romania; silvianajianu@yahoo.com; 6Department of Anatomy and Embryology, Victor Babeș University of Medicine and Pharmacy, E. Murgu Sq., no.2, 300041 Timisoara, Romania; 7Department of Neurology, General Medicine Faculty, Ovidius University, 900527 Constanța, Romania; 8Department of Internal Medicine II-Division of Nephrology, Victor Babeș University of Medicine and Pharmacy, E. Murgu Sq., no.2, 300041 Timișoara, Romania; 9Department of Internal Medicine I, Victor Babeș University of Medicine and Pharmacy, E. Murgu Sq., no.2, 300041 Timișoara, Romania; 10Department of Geriatrics-Rehabilitation, Riviera-Chablais Hospital, 3 Prairie Av, 1800 Vevey, Switzerland; raphaelsadik@hopitalrivierachablais.ch; 11Department of Multidetector Computed Tomography and Magnetic Resonance Imaging, Neuromed Diagnostic Imaging Centre, 300218 Timișoara, Romania; nicoiacob@yahoo.co.uk

**Keywords:** cerebral veins and dural sinuses thrombosis (CVT), thrombophilia, headache, native and contrast-enhanced Head Computed Tomography (CT), Magnetic Resonance Imaging (MRI) of the Head, Magnetic Resonance (MR) Venography

## Abstract

(1) Objective: This review paper aims to discuss multiple aspects of cerebral venous thrombosis (CVT), including epidemiology, etiology, pathophysiology, and clinical presentation. Different neuroimaging methods for diagnosis of CVT, such as computer tomography CT/CT Venography (CTV), and Magnetic Resonance Imaging (MRI)/MR Venography (MRV) will be presented. (2) Methods: A literature analysis using PubMed and the MEDLINE sub-engine was done using the terms: cerebral venous thrombosis, thrombophilia, and imaging. Different studies concerning risk factors, clinical picture, and imaging signs of patients with CVT were examined. (3) Results: At least one risk factor can be identified in 85% of CVT cases. Searching for a thrombophilic state should be realized for patients with CVT who present a high pretest probability of severe thrombophilia. Two pathophysiological mechanisms contribute to their highly variable clinical presentation: augmentation of venular and capillary pressure, and diminution of cerebrospinal fluid absorption. The clinical spectrum of CVT is frequently non-specific and presents a high level of clinical suspicion. Four major syndromes have been described: isolated intracranial hypertension, seizures, focal neurological abnormalities, and encephalopathy. Cavernous sinus thrombosis is the single CVT that presents a characteristic clinical syndrome. Non-enhanced CT (NECT) of the Head is the most frequently performed imaging study in the emergency department. Features of CVT on NECT can be divided into direct signs (demonstration of dense venous clot within a cerebral vein or a cerebral venous sinus), and more frequently indirect signs (such as cerebral edema, or cerebral venous infarct). CVT diagnosis is confirmed with CTV, directly detecting the venous clot as a filling defect, or MRI/MRV, which also realizes a better description of parenchymal abnormalities. (4) Conclusions: CVT is a relatively rare disorder in the general population and is frequently misdiagnosed upon initial examination. The knowledge of wide clinical aspects and imaging signs will be essential in providing a timely diagnosis.

## 1. Introduction

### 1.1. Background

While rare in the adult population, but far more common than previously known, cerebral venous thrombosis (CVT) presents a higher frequency among young adults, children, patients with thrombophilia, pregnant women, puerperium, or women with oral contraceptives therapy [1,2,3].

The clinical spectrum of CVT can be highly variable and usually not specific [4,5,6]. CVT cases rarely present as an arterial stroke syndrome, which is an acute onset of focal neurological deficits associated with classic vascular risk factors [1,2,3].

Because CVT can be produced by multiple predisposing causes and precipitants, it may be encountered not only by neurologists, and neurosurgeons, but also by ear, nose, and throat specialists, ophthalmologists, obstetricians, oncologists, hematologists, rheumatologists, emergency clinicians, family practitioners, and pediatricians [5,6,7]. Different imaging techniques are essential in accurately diagnosing patients with clinically suspected CVT. These are represented by native and contrast-enhanced Head Computed Tomography (CT), CT Venography (CTV), Magnetic Resonance Imaging (MRI) of the Head combined with MR Venography (MRV), and, in peculiar cases, cerebral intra-arterial angiography with venous phase imaging, direct cerebral venography, or ultrasound (US) [3,4,5].

This paper will review the cerebral veins and dural sinuses anatomy, the epidemiology, etiology, pathophysiology, and clinical and imagistic aspects of CVT. The advantages and disadvantages of each imaging method will be analyzed to help the neurologists and radiologists in the selection of the most adequate method/methods to establish a faster and more accurate diagnosis in each peculiar CVT case [5,6,7].

### 1.2. Cerebral Veins and Dural Sinuses Anatomy

Familiarity with the anatomic variants of cerebral veins and dural sinuses is essential to accurately detect CVT.

#### 1.2.1. Cerebral Veins

The cerebral veins are represented by three groups: the superficial cerebral venous system, the deep cerebral venous system, and the posterior fossa veins. (Figure 1) [1,8].

In contrast with other veins, including the internal jugular veins (IJV), the cerebral veins develop certain peculiarities that can explain some of the clinical aspects of CVT. On one hand, the cortical veins, and the posterior fossa veins present wide anatomic variability (in number, location, and anastomoses), thus explaining why the angiographic diagnosis of their isolated occlusion is very difficult. On the other hand, the occlusion of the deep cerebral veins is easy to detect, because these veins (with the exception of the anatomic variations of the basal veins) are constant and are always detected at angiography [1,8].

Cerebral veins have thin walls, without a muscular tunic, possess no valves, and are linked by different anastomoses, which enable both their dilatation and the inversion of venous flow toward the brain if there is an occlusion of the sinus into which they empty. Thus, the presence of anastomoses allows the development of collateral venous circulation in the situation of vessel occlusion, explaining the good prognosis in such patients [1,8].

#### 1.2.2. Intracranial Dural Sinuses

They are divided into two groups: the posterior-superior, and the antero-inferior. (Figure 1) The former group includes the Superior Sagittal Sinus (SSS), Inferior Sagittal Sinus (ISS), Lateral Sinus (LS), consisting of transverse sinus and sigmoid sinus, Straight Sinus (SS), and occipital sinus; the torcular Herophili represents the junction of SSS, SS, transverse sinus, and occipital sinus and is frequently asymmetrical [1,8,9]. The latter group comprises the cavernous sinus, and the superior and inferior petrosal sinuses. The great majority of the cerebral venous blood collects posteriorly, from the SSS or the SS into the LSs, and only a smaller part flows to the cavernous sinuses [1,8].

The intracranial dural sinuses drain in adults into the two IJV, for the supine position, and into the vertebral venous system for the upright position [10].

## 2. Material and Methods

A literature search using PubMed and the MEDLINE sub-engine was completed using the terms: cerebral venous thrombosis, thrombophilia, and imaging. Different studies concerning epidemiology, etiology-risk factors, pathophysiology, clinical diagnosis, laboratory data, and imaging signs of patients with CVT were included.

## 3. Results

### 3.1. Epidemiology

No epidemiologic studies of CVT own the needed criteria for a good quality epidemiologic stroke study, due to multiple factors, including the wide clinical spectrum of CVT, frequently with subacute onset [6,11,12].

CVT represents only less than 1% of all strokes [12], its prevalence is higher than previously noted, due to an increased awareness of this type of disease among clinicians, and to improved and more accessible imaging methods, including MRI/MRV, for the examination of patients with unclear neurological clinical aspects, such as headache and seizures [6,11,12,13,14].

Annual incidence is noted to be between 0.22 to 1.57 per 100,000 citizens [11] and is sex-independent in children and the elderly [6,7]. CVT is more frequent in children (especially in neonates) than in adults [6,7]. In adults, CVT is observed in patients who are younger on average than those with arterial types of stroke [6]. Thus, in the International Study on Cerebral Vein and Dural Sinus Thrombosis (ISCVT) cohort, the median age was 37 years [7], and only 8% of the patients were older than 65 [15]. Compared with men, women were significantly younger (median age of 34 years, vs. 42 years for men) [13]. In adult cases of the ISCVT cohort, CVT presented a female predominance (3:1) [9], which was higher between 31 to 50 years because of augmented risk attributed to a prothrombotic condition [6,7].

### 3.2. Etiology-Risk Factors

The most frequent risk factors detected in elder people with CVT are thrombophilia, neoplasms, and hematologic disorders [15,16].

In the Canadian pediatric ischemic stroke registry, a risk factor was detected in 98% of the children. Thrombophilia was found in 41% of all cases. In infants older than four weeks of age and children head and neck diseases, especially infections and chronic systemic diseases (e.g., connective tissue diseases, hematologic disorders, or malignancy) were frequent [17].

Different factors can determine or predispose adults to develop CVT (in 85% of cases at least one risk factor can be found), and in about half of CVT cases, they present multiple risk factors. For this reason, the detection of a cause or a risk factor should not stop a search for others [7]. The most common risk factors are represented by: genetic thrombophilia, oral contraceptives (OC), pregnancy, puerperium, malignancy, and infections [7,12]. Thus, a prothrombotic condition was detected in one-third of all cases of the ISCVT cohort, and a genetic thrombophilia was reported in 22% of all cases [7].

#### 3.2.1. Genetic Thrombophilia

According to Marjot et al., the risk for CVT depends on the individual’s genetic background [18]. If different prothrombotic aspects are noted, patients present an increased risk of CVT when they are exposed to a precipitant factor such as head trauma, pregnancy, infection, etc. [6].

The major genetic trombophilias as prothrombotic conditions include the following: factor V Leiden (FVL) pathologic variant; [17,19,20] G20210 A prothrombin gene pathologic variant; [17,20,21,22]; hyperhomocysteinemia; [23] antithrombin deficiency; [24] and protein C or protein S deficiencies [16,25].

In a meta-analysis of case-control studies, with more than 200 neonates and children with CVT, and 1200 control subjects, the prevalence of FVL variant among patients with CVT and controls was 12.8% and 3.6%, respectively, and carriers of the FVL variant were significantly more likely to present CVT (odds ratio [OR] 3.1, 95% CI 1.8–5.5) [26]. In the same meta-analysis, the prevalence of the G20210 A prothrombin gene pathologic variant among all patients and controls was 5.2% and 2.5%, respectively, and carriers were significantly more likely to develop CVT (OR 3.1, 95% CI 1.4–6.8) [26].

The association of hyperhomocysteinemia due to different genetic variants in methylene tetrahydrofolate reductase (*MTHFR*) with CVT is controversial [18,21,26]. On one hand, in a meta-analysis of case-control studies, Gouveia and Canhão reported that the frequency of the *MTHFR* 677C > T polymorphism in adults was similar for 382 cases with CVT compared with 1217 controls (15.7% vs. 14.6%; OR 1.12, 95% CI 0.8–1.58), thus indicating that the *MTHFR* 677C > T polymorphism is not a risk factor for CVT [27]. On the other hand, Marjot et al., asserted in a meta-analysis, after controlling for heterogeneity among studies, that the *MTHFR* 677C > T polymorphism was associated with CVT [OR 2.30, 95% CI 1.20–4.42) [18,28].

There is no association of CVT with PAI-1 or protein Z polymorphisms [6,29].

##### Acquired Thrombophilia

According to different studies, the most frequent acquired thrombophilia are due to pregnancy, puerperium, OC, and malignancy [30,31].

#### 3.2.2. Pregnancy and Puerperium

In high-income countries, between 5–20% of all CVT patients present risk factors for pregnancy and puerperium; for example, in the ISCVT cohort, these risk factors appeared in 15% of all CVT patients [7]. In low-income countries, puerperium is the most frequent risk factor for CVT, with 31% of patients [32,33,34,35]. There are multiple favorable conditions, including the absence of antenatal care, home delivery, and depletion of vitamin and protein stores [32,33,34,35]. Usually, CVT appears in the third trimester or, more frequently, in the first three weeks after delivery. In the majority of cases, CVT could be associated with pulmonary embolism and/or lower extremity or pelvic phlebothrombosis, due to the hypercoagulability and the venous stasis that appear during pregnancy [32,33,34,35]. At this stage a significant increase of fibrinogen and different coagulation factors and a notable diminution of antithrombin III and plasminogen occurs. This state of hypercoagulability is accentuated during early puerperium due to multiple causes, including trauma-induced by instrumental delivery or cesarean section, volume depletion, and infections favored by precarious hygienic conditions at home. Pelvic phlebothrombosis may produce CVT via the venous plexuses of the vertebral channel, and the basilar venous plexus [32,33,34,35].

#### 3.2.3. Therapy with Estrogens, such as Hormonal OC or Replacement Therapy

The most common risk factor for CVT in younger women is the use of OC [36,37]. In 10% of such cases, OC represents the only identifiable risk factor [1,7]. In the remaining cases, there are other risk factors for CVT, associated with the OC, such as different vasculitis (systemic lupus erythematous-SLE, Behçet’s disease), obesity, [38] or genetic thrombophilia; in this last situation, the risk of intra-, or extracerebral thrombosis is 6 times higher than that of non-users [7,39].

Different case reports noted the association between tamoxifen (which is an estrogen receptor modulator) and CVT [40].

In contrast to inherited thrombophilia, pregnancy and OC therapy represent transient risk factors for CVT and they are not linked with a higher risk for recurrence [32,33,34,35].

#### 3.2.4. Malignancy Disorders

In the ISCVT cohort, cancers represented 7.4% of all CVT cases [7]. The most frequent malignancy disorders associated with CVT are solid tumors outside the Central Nervous System (CNS) (breast tumors, medullary carcinoma of the thyroid, nephroblastoma, Ewings tumor, gallbladder carcinoma), hematologic malignancies, and CNS malignancies (medulloblastoma) [12]. The main mechanisms are direct tumor compression or invasion of dural sinuses, leukostasis, the hypercoagulable state produced by augmentation in acute-phase reactants, or modified coagulation factors from chemotherapy (L-asparaginase, cisplatin), or hormonal drugs [6,12].

#### 3.2.5. Hematologic Disorders

Different Philadelphia-negative myeloproliferative disorders (MPDs), such as polycythemia vera (PV) or essential thrombocythemia, present an augmented risk of venous thrombosis, including CVT. The acquired Janus kinase 2 V617F mutation (JAK2 V617F), which appears in more than 90% of patients with PV, produces an augmented incidence of severe CVT with a poor prognosis [41]. Other hematological disorders that can produce CVT are represented by paroxysmal nocturnal hemoglobinuria, heparin-induced thrombocytopenia, thrombotic thrombocytopenic purpura, essential thrombocytosis, different gammapathies, iron deficiency anemia, and myelofibrosis [12].

#### 3.2.6. COVID-19 Infection and COVID-19 Vaccine-Associated CVT

##### COVID-19 Infection-Associated CVT

Some cases of CVT have been noted in the setting of SARS-CoV-2 infection, usually without other associated predisposing risk factors [42]. According to the European Medicines Agency safety committee report concerning a review of 34,331 patients hospitalized with SARS-CoV-2 infection, the frequency of CVT was 0.08% (95% CI 0.01–0.5). The in-hospital mortality was 40% [43].

##### COVID-19 Vaccination-Associated CVT

Vaccine-induced immune-mediated thrombocytopenia (VITT) represents an infrequent process of thrombosis associated with thrombocytopenia, frequently producing CVT, and splanchnic veins thrombosis, that has been observed after adenovirus vector vaccines against COVID-19: ChAdOx1 nCOV-19 (AstraZeneca) and Ad26.COV2⋅S Johnson and Johnson (Janssen/J&J) [44]. Although there have been a few reports of CVT after mRNA vaccines, these did not have the features of VITT and could have been incidental [45].

Perry et al. [45] observed in their study that when they are compared with those without VITT, patients with VITT-associated CVT were younger, presented fewer venous thrombosis risk factors, and were more likely to have been administrated the ChAdOx1 vaccine. They presented more extensive CVT with more cerebral veins or dural sinuses thrombosed, and multiple intracerebral hemorrhages (ICH) were more frequent. They were more likely to have concomitant extracranial venous (especially splanchnic veins) or arterial thromboses. Their outcomes at the end of hospital admission were worse, with higher rates of death and disability, ranging from 22% to 47% in studies, compared with 3% to 5% among those with other causes of CVT [45,46].

The diagnostic criteria for definite VITT-associated CVT are post-vaccine CVT (between 4 and 28 days after COVID-19 vaccination), thrombocytopenia (lowest recorded platelet count <150 × 10⁹ per L or documented platelet count), anti-platelet factor four (PF4) antibodies (detected on ELISA or functional assay) [45,46,47].

Other laboratory data are represented by high levels of D-dimer (>4000 mcg/L) and disseminated intravascular coagulation-like coagulopathy with a tendency to hemorrhage [44].

#### 3.2.7. Infections

In the past, loco-regional or systemic infections represented the main cause of CVT. Nowadays, in developed countries (especially due to higher accessibility to antibiotics) septic thrombosis of cerebral veins and dural sinuses in adults has become a rare (6% to 12% of cases), but sometimes severe disease [7,30].

In developing countries, different infections remain an important cause (18% of cases) [34]. Localized acute pyogenic infections of the middle third of the face (especially with Staphylococcus aureus), of the paranasal sinuses, mouth with multiple dental abscesses, ears (otitis media), mastoid air cells (mastoiditis), throat, or scalp can produce septic CVT, especially for the cavernous and lateral sinuses. In chronic CVT, the main identified germs are gram-negative or fungi (especially Aspergillus). Cerebral thrombophlebitis may also appear as a complication of meningitis, epidural, or brain abscesses, or after an open traumatic injury of the head, pelvic phlebothrombosis, or after systemic infectious diseases (trichinosis, human immunodeficiency virus or cytomegalovirus) [1].

#### 3.2.8. Systemic Autoimmune Disorders

The most frequent are SLE, with or without the nephrotic syndrome, Behçet disease, Sjögren’s syndrome, Wegener’s granulomatosis, sarcoidosis, and inflammatory bowel disease [6,12].

#### 3.2.9. Head Injury and Mechanical Precipitants

These are less common causes of CVT. Cerebral veins and dural sinuses could be occluded by different local factors, including head trauma (with local injury to cerebral sinuses or veins), brain tumors, arachnoid cysts, arterio-venous malformations, and by mechanical causes, such as neurosurgical procedures or systemic surgery, epidural blood patch, jugular venous cannulation, spontaneous intracranial hypotension, lumbar puncture [1,48].

#### 3.2.10. No Identified Cause

There is still an important number of idiopathic CVT patients. According to ISCVT cohort data, in 13% of adult CVT cases, no risk factors could be identified [7]. The percentage of idiopathic CVT was higher for older patients (37%) [7,15]. No identified etiology or risk factor for CVT was noted in a minority of children (≤10%) [17].

### 3.3. Pathophysiology

The main two pathophysiological mechanisms implied in CVT are diminution of CSF absorption, and increase of venular and capillary pressure [3,49,50,51] (Figure 2).

#### The Occlusion of the Intracranial Dural Sinuses May Determine a Diminution of CSF Absorption

The normal absorption of CSF is produced through arachnoid granulations, especially at the level of the SSS and LS. In the particular situation of dural sinuses occlusion, especially of the SSS and LS, appears a raise of the cerebral venous pressure, with subsequent diminution of CSF absorption which, consecutively, increases the intracranial pressure. This pathologic process produces a rise in venular and capillary hypertension and generates vasogenic and cytotoxic edema and cerebral hemorrhage [3,49,50,51].

The second mechanism is represented by the progressive increase of venular and capillary pressure.

This mechanism is the result of the thrombosis of dural sinuses and cerebral veins [3,49,50,51].

In the initial stages of venous obstruction, a diminished but still efficient perfusion of the correspondent brain tissue might be possible, due to the collateral venous circulation, which produces a significant degree of compensation, with consecutive neutralization of the pathological pressure modifications. For this reason, the corresponding areas of the brain can be functionally and metabolically affected, but not irreversibly injured [3,49,50,51]. As loco-regional venous pressure continues to increase, in the context of an ineffective collateral venous circulation, a rise of the thrombosis within cortical venous tributaries will reduce the cerebral perfusion pressure even more. Consequently, it will appear damage to the blood-brain barrier producing vasogenic edema, local ischemic lesions, cytotoxic edema, and venous and capillary lesions with consecutively parenchymal hemorrhage and, rarely, subarachnoid hemorrhages [3,49,50,51].

The diminution of the venous drainage consecutive to CVT determines raised venous pressure, with the backup of the fluid into the brain, producing vasogenic edema. This type of edema is situated within the extracellular compartment of the encephalic white matter/inside the glial cells, under the control of the hydrostatic pressure (augmented blood pressure and local blood flow) and osmotic gradients. Usually, the vasogenic edema does not produce neuronal lesions, because the fluid in excess in the extracellular space can, frequently, be removed [3,49,50,51].

Cytotoxic edema is produced by energy failure with a displacement of ions and water across the cell membranes into neurons. The intracellular edema, caused by ischemia, determines a great volume of dead or dying brain neurons with a bad prognosis [3,49,50,51].

In cerebral venous infarcts, the vasogenic edema represents the majority in comparison with the cytotoxic edema; these pathological aspects identified by diffusion-weighted imaging (DWI) confirm that the venous infarcts differ from arterial ones and present a significantly better recovery [1,49,50,51].

Brain edema and associated augmented intracranial pressure produce headache, vomiting, and diminished consciousness, but the most severe complications are represented by the pressure differences and the potential risk for brain herniation, which can determine death due to probable pressure-related lesions to neighboring areas [1].

The growth of the venous and capillary pressures produces vessel lesions and erythrocytes diapedesis due to disruptions of the blood-brain barrier both resulting in cerebral hemorrhage. The neuronal lesions determined by the cerebral hemorrhage induced by CVT are often milder than the damages induced by arterial infarcts [3,49,50,51].

Histological exam in CVT cases notes dilated cerebral veins, brain edema with flattened gyri, diminished sulci, compressed small ventricles, and ischemic neuronal lesions. The thrombus inside the cerebral veins is similar to other venous thrombi (when it is fresh, it presents a rich content in red blood cells and fibrin and a poor content in platelets; and, when it is chronic, it is replaced by fibrous tissue, frequently with recanalization) [1].

### 3.4. Clinical Diagnosis

The clinical spectrum and outcome of CVT are related to different factors: location and number of thrombosed sinuses and cerebral veins, as well as the presence of functional collateral pathways, absence or presence of parenchymal lesions (cytotoxic or vasogenic edema, hemorrhage), gender, age, etiology, and interval from onset to admission to hospital [4,6].

The clinical presentation of CVT can be polymorphous, and misleading. In the majority of cases (50–80% of patients), the onset is subacute [1,52].

#### 3.4.1. Clinical Syndromes

In the majority of adult CVT cases, four major clinical syndromes have been noted in combination or isolation: Isolated intracranial hypertension, focal neurological deficits, seizures, and encephalopathy [1,2].

A minority of adult CVT patients develop a cavernous sinus thrombosis with a distinctive clinical picture: painful ophthalmoplegia. Collet-Siquard syndrome (consisting of multiple low cranial nerves palsies) represents a clinical syndrome of IJVs, posterior fossa veins, or LS thrombosis [1,2].

Rare adult CVT cases with unusual clinical aspects were also reported: subarachnoid hemorrhage, transient ischemic attacks, or psychiatric symptoms, mimicking a postpartum psychosis [1,2].

In neonates, CVT has a nonspecific clinical presentation with seizures, tetraparesis, and encephalopathy [17]. In older children, the clinical spectrum is more similar to the adult clinical aspects, with headache and paresis [53]. In elderly patients, symptoms of encephalopathy are more common than in adults, whereas isolated intracranial hypertension is less common [1,2].

##### Isolated Intracranial Hypertension

It is the most frequent clinical syndrome observed in CVT (40% of cases) [6]. It consists of headaches, associated with vomiting, papilledema, visual complaints, and sixth nerve palsy [54]. It is more common in patients with a chronic onset than in those cases that present acutely [55].

Headache is the most common symptom of CVT (about 90% of cases in the ISCVT cohort). It is usually the initial one, and can develop isolated, or can precede other symptoms or signs. Headache is more frequent in women and younger patients than in men or older patients [7,13,52,53]. The characteristics of headaches are polymorphic. It may be localized or diffused [54]. Frequently, headache is severe augmenting during the night and may worsen with Valsalva maneuvers or position changes (when the patient is lying down) [2,32].

However, its characteristics can be misleading, sometimes being initially diagnosed as a migraine with aura [56]. In a few cases, it occurs like a thunderclap headache (mimicking a subarachnoid hemorrhage) [57]. Some of the risk factors associated with CVT (such as meningitis, epidural or brain abscesses, meningiomas, dural arteriovenous fistulas, and different vasculitis) also clinically manifest as a headache. CVT must be suspected as a possible explanation of persisting headache after lumbar puncture because this maneuver can rarely precipitate a CVT [1]. Headache is noted more frequently in patients with CVT than in cases with cerebral arterial infarcts [1,6].

Papilledema is observed on funduscopy in 25–40% of CVT cases, especially in those with chronic onset or delayed clinical presentation. It can produce transient loss of vision (associated with intense headache), and if prolonged, optic atrophy and consecutive peripheral blindness [6,12].

##### Focal Neurological Deficits

They are noted in 37–50% of CVT patients and appear at onset in 15% of cases [1,7].

Paresis, sometimes bilateral, is the most frequent focal neurological deficit associated with CVT (in the ISCVT cohort was noted in 37% of cases) [1,7].

Other signs are less common: fluent aphasia (which is observed in left transverse sinus thrombosis associated with a posterior left temporal lesion), central sensory deficits, hemianopia, and ataxia (usually observed in posterior dural sinuses occlusion) [7]. Mixed transcortical aphasia is noted in left thalamus lesions due to deep cerebral vein thrombosis [58].

##### Seizures

Focal or generalized seizures, even status epilepticus, are more frequently noted during the evolution of CVT (in the ICSVT cohort in 40% of cases) [7] than in arterial strokes [59,60,61]. Seizures appear during the onset of CVT in about 12–15% of cases [59,60]. A higher incidence has been observed in peripartum (76%) [60] and neonates (44%) [61].

Early seizures are noted more frequently in cases with supratentorial parenchymal brain lesions (especially disposed anterior to the sulcus of Rolando), thrombosis of the SSS and cortical veins and in those patients who present motor deficits [59,60,61].

##### Encephalopathy

Subacute/chronic encephalopathy, presenting altered mental status with cognitive dysfunction (including delirium, apathy, and dysexecutive syndrome), and diminished level of consciousness (between drowsiness and deep coma) is frequently associated with multifocal neurological deficits and is observed especially in elderly patients or neonates with CVT [15,62]. Usually, the diminution of the level of consciousness is reversible; however, coma at onset represents the main predictor of a poor outcome [1,2].

#### 3.4.2. Topographic Clinical Diagnosis

Due to frequent concomitant multiple cerebral veins and dural sinuses thrombosis (more than two-thirds of cases), the existence of multiple anatomic variants of some cerebral veins and dural sinuses, and action of venous collateral circulation, the topographic clinical diagnosis of CVT is not so well-defined like in arterial occlusion and frequent is misleading [1,7,63].

However, isolated thrombosis of the different dural sinuses and cerebral veins produces the following clinical aspects (Figure 1).

##### Superior Sagittal Sinus (SSS) Thrombosis

It represents the most frequent dural sinuses occlusion, especially during the puerperium (62–80% in association, and 30% in isolated thrombosis) [1,6,12]. The common clinical presentation is that of an isolated intracranial hypertension syndrome. The clinical aspects may vary depending on the concomitant occlusion of other dural sinuses, especially LSs cerebral, or tributaries cerebral veins. Bilateral motor/sensory signs (especially in the legs) and psychiatric symptoms (prefrontal syndrome) may also appear due to bilateral frontoparietal hemispheric lesions produced by the progression of the SSS thrombosis to tributaries bilateral cortical veins [1,6,12].

##### Lateral Sinus (LS) Thrombosis

LS thrombosis may present different clinical aspects. Cases with isolated LS thrombosis develop an isolated headache or frequently intracranial hypertension (pseudotumor) syndrome. Less often, these patients may also present with focal neurological deficits due to hemispheric lesions produced by the progression of the LS thrombosis to tributaries cortical veins.

In contrast to SSS thrombosis, the infectious etiology is much more common in LS thrombosis. Different localized pyogenic infections of the ears (otitis), mastoid air cells (mastoiditis), and the paranasal sinuses (sinusitis) can determine septic LS thrombosis: “otitic hydrocephalus” [1,63]. The clinical signs are relatively characteristic: fever, headache, neck pain, neck tenderness, nausea and vomiting, vertigo, diplopia produced by sixth nerve palsy, and temporal and retro-orbital pain due to symptomatic trigeminal neuralgia [1,12,63].

Since the left LS is often hypoplasic, the pseudotumor syndrome appears especially after right LS thrombosis. In such cases, a bilateral venous drainage impairment may be noted, affecting the inferior portions of both temporal lobes and cerebellum, with subsequent temporal lobe and cerebellar signs [1,2,63]. Fluent Wernicke aphasia is usually observed in left transverse sinus thrombosis associated with adjacent cerebral veins occlusion (40%) and can be associated with right hemianopia or superior quadrantanopia. Right temporal lobe lesions produce left hemianopia. Nystagmus and gait ataxia represent the markers of cerebellar affection [1,7,63].

Unusually, an isolated left LS thrombosis presents a misleading isolated headache (migraine-like). In such cases, the thrombosis is not due to an otitis, but a thrombophilia [3,63,64]. This is why screening for LS thrombosis (and other CVT) has to be done in young women with a recent headache even if this symptom is isolated and is not associated with otitis or mastoiditis [63,64]. LS thrombosis may present also as isolated pulsating tinnitus [65].

##### Cavernous Sinus Thrombosis

It is rare and usually has an infection etiology (pyogenic infections of the face, or the paranasal sinuses) [66,67].

In patients with classic acute unilateral septic cavernous sinus thrombosis, they present a typical clinical picture, with painful complete or partial ophthalmoplegia associated with chemosis, proptosis, and conjunctival edema. Frequently, a papilledema associated with hemorrhages of the retina can be observed. In the absence of an immediate diagnosis and treatment, it becomes bilateral via inter-cavernous sinuses. When the thrombosis progresses to other dural sinuses and cortical veins, seizures and paresis may associate [66,67].

In a minority of cases (head trauma, surgery on intracranial or facial structures, thrombophilia, and thrombosis of dural arteriovenous fistulas) an aseptic cavernous sinus thrombosis may be observed with an isolated abducens nerve palsy and mild proptosis [66].

##### Thrombosis of the Superior and Inferior Petrosal Sinuses

In the majority of cases, it represents a sequela of cavernous or sigmoid occlusion. The thrombosis of the superior petrosal sinus clinically manifests as a trigeminal palsy, while the occlusion of the inferior one occurs as an abducens palsy [1,2].

##### Cortical Veins Thrombosis

Isolated thrombosis of cortical veins without associated dural sinus thrombosis is considered a rare disease (2%), but it is probably underdiagnosed, due to difficulties to detect it using the traditional MRI sequences and MRV [68]. Occlusion appears especially at the levels of the superior cortical veins, producing seizures associated with motor/sensory deficits [68].

##### Thrombosis of the Deep Cerebral Venous System

Frequently associated with the thrombosis of the SS, it is a rare disease that occurs more often during childhood (especially in neonates). The clinical presentation is severe, with encephalopathy, and bilateral motor deficits [69,70].

In adults, a more limited thrombosis of the deep cerebral veins, without associated SS thrombosis, can produce relatively mild clinical aspects, such as headache, nausea and vomiting, gait ataxia, hemiparesis (that may be bilateral or alternating), neuropsychological symptoms, and mild disturbances of consciousness [69,70]. In rare situations, benign cases of thrombosis of the deep cerebral veins were noted with mixed transcortical aphasia [58].

##### Thrombosis of the Posterior Fossa Veins

Isolated venous infarctions in the posterior fossa are rare because the posterior fossa owns an efficient collateral venous circulation. However, this disease represents the main differential diagnosis in cases with concomitant risk factors for CVT, associated with some clinical aspects (intracranial hypertension syndrome, cerebellar-vestibular syndrome), and atypical aspects on brain CT (bilateral hemispheric and vermian cerebellar infarcts, irregular cerebellar hemorrhages) [71,72].

##### Internal Jugular Vein (IJV) Thrombosis

IJV thrombosis represents especially a progression of the sigmoid sinus thrombosis or may be produced by cannulation for long-term IJV access. It can be asymptomatic or its clinical picture consists of symptoms and signs of local infection (such as pain and swelling of the mastoid area, a painful and tender thrombosed IJV). The thrombophlebitis of the IJV can be a consequence of the syndrome of tonsillopharyngitis (Lemierre’s syndrome). A jugular foramen syndrome (consisting of unilateral pulsatile tinnitus [65] or multiple cranial nerve palsies VIII-XII) occurs if the infection affects the skull base [73].

##### Emissary Veins (EV) Thrombosis

The EVs (e.g., petrosquamosal sinus (PSS)), are vestigial valve-less veins, which connect the intracranial dural venous sinuses and the extracranial venous system. Posterior fossa EVs pass through corresponding cranial apertures and ensure (together with the IJV) an additional extracranial venous drainage of the veins of the posterior fossa. In healthy people, EVs are small and have no clinical significance. In pathological cases (associated with high-flow arteriovenous malformations, IJV aplasia or IJV thrombosis, or even LS thrombosis) they are large with clinical significance (different craniofacial syndromes and pulsatile tinnitus) [65,72].

### 3.5. Laboratory Data

Unfortunately, there is no simple confirmatory laboratory test that can rule out an acute CVT.

#### 3.5.1. Blood Assay

Guidelines from the American Heart Association/American Stroke Association (AHA/ASA) recommend obtaining a complete blood count, chemistry panel, prothrombin time and activated partial thromboplastin time for cases with clinically suspected CVT [12]. These data may reveal conditions that contribute to the appearance of CVT such as a hypercoagulable state, infective, or inflammatory, diseases.

Anti-platelet factor four (PF4) antibodies are searched for COVID 19-vaccination-associated CVT [45]. An assessment for paroxysmal nocturnal hemoglobinuria should be done if the complete blood count indicates hemolytic anemia, iron deficiency, or pancytopenia [6].

The guidelines recommend screening for use of oral contraceptives, at the initial clinical presentation of young women, and for an occult neoplasm or hematologic disorder in patients older than 40 years with suspected CVT [12,74].

#### 3.5.2. D-Dimer

An elevated plasma D-dimer level suggests the diagnosis of CVT, but a normal D-dimer level does not exclude the CVT diagnosis, especially in those cases with predisposing factors and a compatible clinical presentation for CVT, such as isolated headache or in those patients with a subacute or chronic clinical presentation before the D-dimer test [75,76].

#### 3.5.3. Lumbar Puncture and Cerebrospinal Fluid (CSF) Examination

They may be useful to exclude meningitis in cases with CVT who clinically present with isolated intracranial hypertension, and in cases with sepsis, or with fever and no obvious cause of infection [7,12,77]. Increased opening pressure during the exam of CSF pressure represents a frequent feature in CVT cases presenting isolated intracranial hypertension. Other significant data are nonspecific and may be represented by a mild lymphocytosis, hyperproteinorahia, and numerous red blood cell count; these abnormalities are noted in 30% to 50% of cases with CVT [77].

Lumbar puncture is contraindicated in cases with CVT and with large brain lesions because they present an augmented risk of herniation [6,77].

#### 3.5.4. Evaluation for Thrombophilic State

Searching for a thrombophilia should be realized for cases with CVT who present an elevated pretest probability of severe thrombophilia (patients with a personal and/or family history of venous thrombosis, CVT in young people, or CVT in the absence of a transient or permanent risk factor) [6].

When indicated, screening should contain factor V Leiden, prothrombin G20210A pathologic variant, antithrombin, protein C, protein S, lupus anticoagulant, anticardiolipin, and anti-beta2 glycoprotein-I antibodies [6,7,74].

### 3.6. Imaging

Neuroimaging data are essential for the diagnosis of CVT [74,78].

#### 3.6.1. Head Computed Tomography (CT)

Head CT is usually the first technique to be performed in the emergency department in cases with acute/subacute clinical suspicion for CVT [12]. It should be done initially native (NCECT), and then (if hemorrhagic infarcts, cerebral hemorrhage, subdural hematoma, or subarachnoid hemorrhage are absent) with contrast enhancement (CECT) [12,74,78] (Table 1).

Head CT presents the following advantages:

First, it may identify alternative diagnoses that CVT can clinically resemble, such as subdural hematoma, abscess, or different tumors. Second, sometimes, CT diagnoses diseases that can themselves produce CVT, such as sinusitis, mastoiditis, abscesses, or meningiomas. Third, Head CT can detect direct and indirect signs of CVT [1,74,78].

##### Direct Signs of CVT on Head CT

They represent the direct visualization of the venous clot inside the occluded cerebral vein, or dural sinus, and can be observed in about one-third of all CVT patients. They are the “cord sign”, the “dense triangle sign,” and the “empty delta sign” [6,74,79,80]. (Figure 3) [58].

The “cord sign” represents an acute thrombosed cerebral vein on NCECT and is reported in a quarter of CVT cases. It appears as a curvilinear or linear hyperdensity representing a fresh clot in-side an occluded cerebral vein and it is best visualized within the first 7 days of CVT clinical onset [8]. After 1 to 2 weeks, the thrombus appears isodense and then hypodense. Mimicking appears in slow flow cases; thus, its specificity is considered to be rather low [1,74,78].

The “dense triangle sign” (acute thrombus in dural sinus on NCECT) is identified in less than 2% of CVT patients [78]. It appears as a hyperdensity with a triangular or round shape in the posterior part of the SSS caused by the fresh clot inside the sinus [6]. It can be detected during the first 2 weeks of the disease. Because cases with increased hematocrit or dehydration can also produce this sign, its specificity is considered to be rather low, especially in other sinuses than SSS [79,80,81]. Measurement of the venous sinus density and Hounsfield unit-to-hematocrit (H:H) ratio has been noted to raise the sensitivity in detecting CVT, as attenuation of 62HU and higher is suggestive of thrombosis [82]. If a hyperdense sinus is detected, consecutive investigation with CTV and/or MRI/MRV should be realized [1,2].

The “empty delta sign” is detected on CECT scans in 10–20% of CVT cases. It appears as a triangular hyperdensity of contrast enhancement of the walls of the dural sinus surrounding a hypodense central region lacking contrast enhancement inside the posterior part of the SSS [6,83,84]. This sign occurs between day 5 to 2 months after onset. The sensitivity and specificity of the “empty delta sign” are augmented to 30% of all CVT cases with CT scans with orthogonal sectioning, different window and level settings, and multi-planar reformations. Furthermore, an early division of the SSS can mimic a false “empty delta sign”, so this sign is not pathognomonic [83,84].

##### Indirect Signs of CVT on Head CT

They are more frequent than direct signs and can include intense contrast enhancement of falx and tentorium, dilated cerebral veins, small or enlarged ventricles, and cerebral parenchyma abnormalities [7,12].

Intense contrast enhancement of falx and tentorium represents venous stasis or hyperemia of the cranial dura mater and is detected in 20% of cases. The former can be difficult to examine, especially in older patients, but the latter is more easily identified, denoting especially SS and, sometimes, SSS thrombosis [1,7,83].

The cortical veins may appear dilated on the CECT exam due to collaterally veins development [83].

Diffuse brain edema (20–50% of CVT cases) can secondarily produce effacement of cerebral sulci and small ventricles (diminution of the ventricular dimensions); this imaging sign may be difficult to differentiate from normal aspects with small ventricles in young adults [7,83].

The identification of the opposite sign (enlarged ventricles) cannot remove the diagnosis of CVT, because it may be produced by the hydrocephalus appearing from raised CSF production and diminished resorption from augmented venous pressure; usually, it can be associated with posterior fossa veins occlusion. In both cases (small, vs. enlarged ventricles), a comparison with precedent CT exams may be necessary [7,83].

Cerebral parenchyma abnormalities may be nonhemorrhagic or hemorrhagic and may be detected in 60–80% of CVT cases [83].

The first type includes focal areas of hypodensity produced by edema or venous infarction, usually not respecting the arterial boundaries, as well as diffuse brain edema. With serial imaging, some of them may disappear (“vanishing infarcts”), and new lesions may be detected. The latter type includes hemorrhagic infarcts, intracerebral hemorrhage, or rarely (<1%) subarachnoid hemorrhage usually limited to convexity [6,83].

Peculiar forms of CVT may present on CT different aspects. First, many irregular filling defects with enlarged cavernous sinuses and orbital veins show up on CECT of cavernous sinus thrombosis; in septic cavernous sinus thrombosis, air can be detected inside the sinus on coronal sections [12,74]. Second, bilateral parasagittal hemispheric lesions are highly suggestive of thrombosis of the SSS [12,74]. Third, temporo-occipital lesions indicate LS thrombosis or occlusion of the vein of Labbe [12,74]. Forth, in cases of deep cerebral veins thrombosis, the main features are represented by: bilateral hypodensities, denoting infarcts at the level of the thalami, basal ganglia, and internal capsule; bilateral hyperdensities (hemorrhages or hemorrhagic infarcts) with the same topography; severe edema with compression of the third ventricle and consecutive hydrocephalus, and hyperdense appearance (fresh clot) at the level of the occluded sinuses. Thus, the appearance of hemorrhage or edema near a main dural sinus or deep cerebral vein should suggest CVT [12,74]. Last, cerebellar venous infarctions can determine hydrocephalus and compression of the fourth ventricle [12,74].

Unfortunately, head CT diagnosis of CVT is insensitive, findings being pathologic only in one-third of real CVT cases, and different CT signs are mostly nonspecific in the rest of detected CVT cases [12]. Furthermore, several normal anatomic variants may mimic sinus thrombosis, including sinus atresia, sinus hypoplasia, asymmetric sinus drainage, and normal sinus filling defects associated with arachnoid granulations or intrasinus septa [7,12]. For this reason, normal Head CT results will not exclude a diagnosis of CVT; therefore, in clinically suspected cases, a CT Venography or MRI Venography is essential for CVT positive diagnosis [79,80].

#### 3.6.2. CT Venography-CTV (Multi-Detector CT Angiography-MDCTA) with Bolus Injection of Contrast Material

When associated with head CT, it adds important information in suspected clinical cases of CVT [6]. The combined accuracy of head CT and CTV is 90–100%, depending on the occlusion site [85,86,87,88]. According to AHA/ASA guidelines published in 2011, CTV is considered equivalent to MRV in the diagnosis of CVT [12]. Compared with digital subtraction intra-arterial angiography, the combination of head CT and CTV presents sensitivity and specificity of 95 and 91% [12] (Table 2).

This method is used especially in acute cases, in the emergency department, when it can be performed immediately after NCECT. It ensures an excellent visualization of the cerebral venous system and can detect filling defects in the thrombosed sinuses or cerebral veins, sinus wall enhancement, and augmented collateral venous circulation. In addition, in subacute or chronic CVT phases, it can identify a heterogeneous clot [6,83].

It presents some advantages compared to intra-arterial angiography: it is less expensive and less invasive, and faster (rapid image acquisition); it detects better the inferior sagittal and cavernous sinuses, and the basal vein of Rosenthal (with multiplanar reformatted images) than conventional intra-arterial angiography [85,86,87,88].

The main advantages of CTV vs. MRV are: it is much more accessible and less expensive, faster, has no contraindications to ferromagnetic devices, augmented imaging resolution for the major dural sinuses, much easier to interpret, with fewer artifacts than MRV. MDCTA has been demonstrated to be as accurate as time-of-flight (TOF) MRV in the identification of the dural sinuses, with superior capability vs. MRV to visualize: the ISS and the non-dominant transverse sinus thrombosis; the cortical vein thrombosis (especially for single cortical vein occlusion); and cerebral veins or dural sinuses with low flow [85,86,87,88].

Unfortunately, CTV is less sensitive in the assessment of the deep venous system and cortical veins. This disadvantage can be ameliorated by realizing multiplanar reformations, which increases the sensitivity of CTV beyond angiography [85,86]. Maximum intensity projection (MIP) image generation occasionally presents diminished detection of skull base components in three-dimensional display, with inadvertent sinus exclusion from bone subtracting algorithms; however, this can be improved with specific software for mask bone elimination [88]. CTV also presents different concerns (contrast allergy, contrast nephropathy due to contrast material, and radiation exposure), which may contraindicate its use during pregnancy, childhood, or renal failure [88].

#### 3.6.3. Magnetic Resonance Imaging (MRI) of the Head

MRI pathological features in CVT cases are represented by direct signs (visualizing the clot itself inside the cerebral veins and dural sinuses) and indirect signs (detected especially at the level of the brain parenchyma) [12,89,90] (Table 1).

The MRI direct signs:

The main characteristic of thrombosis is the absence of a flow void or a flow signal with altered intensity in the dural sinus or cerebral vein. The signal intensity of the thrombus on T1-, and T2-weighted MR images is similar to a hematoma, and it is evolving depending on clot age. The successive intensity modifications noted in the clot are the results of the paramagnetic features of the hemoglobin molecule and its secondary products [12,89,90,91,92,93,94].

Acutely (0–5 days after the onset of the clinical aspects), flow void is missing and the thrombosed dural sinus or cerebral vein is isointense with brain tissue on T1-WI and hypointense on T2-WI, due to the abundance of deoxyhemoglobin in red cells within the clot. The detection of CVT in this phase is practically very difficult on MRI alone because the MRI aspect is similar to normal flow. For this reason, other MRI sequences, MRV, CTV, or cerebral intra-arterial angiography are needed to certify the absence of flow in the occluded venous channel [12,89,90] (Figure 4) [58].

In the subacute phase (between days 6 and 15 after the onset), the venous clot becomes more apparent because the signal is hyperintense in both T1-, and T2-WI, due to the accumulation of methemoglobin inside the venous thrombus. This imaging pattern certifies the aging of the clot, rather than its progression. Practically, these intermediate imaging aspects (increased signals on both T1-, and T2-WI) are specific for CVT and are the most common imagistic sign [89,90].

In the chronic stage (between two and four weeks after onset), the beginning of recanalization of the anteriorly occluded cerebral vein or dural sinus determines the recommencing of the flow void. In this phase, the clot, which is heterogeneous, is isointense on T1-WI, with variable intensity (iso-, or hyperintense) on T2-WI, due to the deoxygenated hemoglobin and methemoglobin components. In consequence, in this phase, the diagnosis of CVT can be overlooked [89,90].

After 4–6 months, no signal abnormality is seen on T1-WI or DWI; however, subtle changes (heterogeneous topic signal abnormalities) can be noted in T2-WI or FLAIR, which can remain for years, and should not be confused for a recurrent acute CVT [89,90].

Unfortunately, in a significant number of situations, we can detect false-negative or false-positive appearances. The former situations are relatively rare and represent a supra-acute or very late phase of CVT, or an isolated cortical vein thrombosis, which will be diagnosed especially by angiography. The later situations are the result of slowly flowing venous blood. To reduce these artifacts, we have to reposition the patient, repeat the sequence in a different plane, using at least two sequences, and using special acquisitions, such as [74,83].

Gradient echo T2*-weighted (T2*GRE) MRI sequences identify CVT, as deposited blood breakdown products (i.e., hemosiderin, methemoglobin, deoxyhemoglobin), which can produce increased signal drop-out, detecting intravessel clots in stages where the thrombus can be unapparent in other sequences [91]. Thus, on T2*GRE MRI sequences, the acute clot can be directly detected as an area of hypointensity in the affected cortical vein and/or dural sinus. However, a chronically thrombosed sinus may still have a low signal on these sequences [92].

Echo-planar T2 susceptibility-weighted imaging (T2* SWI) MRI sequences represent a complementary T2* GRE sequence to evaluate CVT. SWI identifies the isolated cortical vein thrombosis during the first days of acute CVT when T1 and T2 are less sensitive. This imagistic method detects the intraluminal clot as a hypointense area. The exaggeration of magnetic susceptibility effect (MSE) helps detection of discrete thrombosis (as it is noted in more than 90% of cases), and it also detects venous stasis, presence of collateral venous pathways, and possible associated intracranial hemorrhage [93]. Supplementary, SWI indicates a blooming artifact better observed than regular T2* GRE sequences, realizing a better location of the thrombus or hemorrhage. Isolated cortical vein thrombosis may be easier to observe on the maximum-intensity projections (MIPs) of SWI compared to dedicate venous imaging [80,93].

The MRI indirect signs:

A variety of parenchymal brain lesions secondary to venous occlusion (brain edema, cerebral infarct, and/or cerebral hemorrhage) noted in CVT cases are better detected by MRI, than by CT [79].

Cerebral edema consists of an increased signal on T2-WI and an isointense or hypointense signal on T1-WI. Isolated cerebral edema, without cerebral infarcts or hemorrhages, may appear in 50% of CVT patients, and may be associated with cortical sulcal effacement and small ventricles. When these MRI signs are detected, CTV or MRV should be done to confirm the CVT diagnosis [89,90].

An augmented signal in both T1- and T2-WI is a hallmark of cerebral hemorrhage (one-third of the CVT patients). Usually, in the case of an SSS occlusion, we can note flame-shaped, irregular, and heterogeneous areas of lobar hemorrhages in the parasagittal areas of both frontal and parietal lobes. Frequently, LS thrombosis is associated with both temporal and occipital lobes abnormalities. Deep brain lesions, especially bilateral thalamic hemorrhages, extensive brain edema, or intraventricular hemorrhage indicate an occlusion of the vein of Galen or SS [12,89,90].

All the MRI indirect signs are nonspecific, but their interpretation is clear because of the associated MRI direct signs of dural sinus thrombosis [89,90].

After contrast (gadolinium) injection, intense contrast enhancement and flow voids may be noted within the occluded sinuses, diminished flow in sinuses and intra-clot collateral channels, or recanalization of the thrombus [1,7].

##### Diffusion-Weighted Imaging (DWI) Techniques

DWI directly identifies the clot as a high signal intensity within the occluded vessel, with decreased apparent diffusion coefficient (ADC). Cases with restrictions on DWI present longer recovery time and a lower probability of total thrombus recanalization (DWI-prognostic factor) [12,94].

DWI also detects cerebral edema, which can be differentiated in:-Vasogenic edema, which is represented by different signal abnormalities in the affected region, and augmented ADC values, practically without significantly lower ADC values than in unaffected areas [1,94].-Cytotoxic edema, which is represented by high signal intensity, and low ADC values [1,94].

On perfusion-weighted (PWI) MRI, relative cerebral blood volume (rCBV), and mean transit time (MTT) are increased in affected regions, with preserved relative cerebral blood flow (rCBF) [1,94].

Due to the prominence of vasogenic edema vs. cytotoxic edema in CVT cases, corresponding regions of the brain may be functionally and metabolically affected, but not irreversibly. The reversibility of different cerebral venous lesions is characteristic of CVT cases and is manifested in both a better recovery in venous infarcts than in arterial strokes and vanishing lesions on MRI [89,90,94].

The main MRI advantages vs. CT in CVT diagnosis are represented by sensitivity to venous flow, better capacity to identify the clot itself, and easily repeatable and noninvasive. However, conventional MRI presents limitations, such in comatose patients or in cases of isolated cortical vein thrombosis, (when intra-arterial angiography is mandatory to confirm the diagnosis of CVT) [90]. (Table 1).

#### 3.6.4. Magnetic Resonance Venography (MRV)

MRV can detect the thrombus and can assess the spontaneous recanalization or recanalization following therapy. Total recanalization is not mandatory for symptomatic recovery, and the presence of collateral pathways may also be identified in MRV [95,96]. (Table 2) (Figure 5) [81].

MRV may be done without contrast enhancement (using the TOF technique or phase contrast technique) or with a contrast-enhanced technique [95,96].

The two-dimensional (2D-TOF) method (with 1.5, and 3-mm thick slices in the coronal and axial planes) represents the most frequent technique and is essential in pregnant or breastfeeding women, as well as in case of severe renal failure, where contrast enhancement is prohibited. It detects the absence of flow signal (absence of opacification) of dural sinuses in the case of an occlusion of the vessel, though interpretation can be confounded by normal anatomic variants such as sinus hypoplasia and asymmetric flow [12,95,96].

The indirect signs of CVT include delayed emptying, collateral venous circulation, dilated veins, and tortuous collateral cortical veins (corkscrew veins) [2]. The 2D-TOF technique is superior to its 3D counterpart due to a relative lack of saturation effects and superior sensitivity in the setting of slow venous flow but has a low sensitivity to small vessels with a slow flow [95,96].

Other MR techniques may be useful to differentiate sinus hypoplasia from venous thrombosis. Contrast-enhanced MRV can ensure better detection of cerebral venous vessels, and T2*GRE or T2*SWI MRI sequences will detect normal signals in a hypoplastic sinus and abnormally low signals in the presence of a clot [91,92,93]. A chronically thrombosed hypoplastic sinus will indicate the absence of flow on 2D-TOF MRV and enhancement on contrast-enhanced MRI and MRV [6]. MRV with gadolinium realizes a direct examination of cerebral veins and dural sinuses filling similar to that of CTV, with comparable sensitivity and specificity. Supplementary, both CTV and contrast-enhanced MRV are superior to the TOF and phase-contrast methods, due to different artifacts in these sequences [83].

Unfortunately, conventional MRV presents some limitations: it has a reduced role in detecting: cortical veins and cavernous sinus occlusions, partial thrombosis of other cerebral veins and dural sinuses, or net differentiation between hypoplasia and occluded sinus [6]. A major disadvantage of MRV compared to CTV is that acquisition times are long, and motion artifacts may appear in comatose patients (Table 2) [83,95,96].

Contrast-enhanced MRV with elliptic centric ordering is a newer method, which realizes better examination of smaller cerebral veins and dural sinuses, in which the paramagnetic effect of gadolinium shorts T1 and determines positive intravascular contrast enhancement [83,96]. 3D phase-contrast (PC) MRV presents a better capacity to identify slow flow and may better differentiate between slow flow and thrombus [95,96]. Static contrast-enhanced 3D MRV detects better the intracranial venous system; unfortunately, it may present some limitations in chronic dural sinus thrombosis as the thrombus may be enhanced, miming a patent dural sinus [95,96]. This problem is solved with time-resolved 3D MRV (4D MRV), which produces images with different delays for better detection of the venous thrombus [95]. 4D MRV presents better sensitivity to diagnose CVT than T2-WI, T2*GRE, and TOF-MRV; supplementary, it has better specificity than TOF- MRV, and it identifies chronic CVT [95].

MRI using gradient-echo T2* SW sequences in combination with MRV is the most sensitive imaging technique for detecting the clot and the occluded dural sinus or cerebral vein [92,95,96].

Unfortunately, the interobserver agreement on CVT MRI diagnostic imaging is not perfect, percentages vary depending on which sinuses or veins are occluded: it is good or very good for most of the occluded sinus and veins; moderate to very good for the left LS and SS; and poor to good for the cortical veins [6,89].

#### 3.6.5. Cerebral Intra-Arterial Angiography with Venous Phase Imaging and Direct Cerebral Venography

These are invasive diagnostic methods that are used for the peculiar cases when: The clinical suspicion of CVT is high, but MR or CT Venography data are ambiguous (such as in cases of isolated cortical vein thrombosis), or when required to exclude a dural arteriovenous fistula or distal aneurysm, like in subarachnoid hemorrhage cases [12,97].

##### Cerebral Intra-Arterial Angiography with Venous Phase Imaging

It requires a four-vessel angiography (conventional or digitized intra-arterial) with detection of the whole venous phase on at least two projections (frontal and lateral) and three oblique views for the detection of the whole SSS [12,97].

Characteristic imagistic signs of CVT on angiography are: partial lack of opacification or absence of filling of dural sinuses or cerebral veins delayed emptying, dilatation of cortical, scalp, or facial veins, dilatation of collateral veins, reversal of venous flow, and the sudden stopping of cortical veins surrounded by tortuous and dilated collateral “corkscrew” veins [12,97]. CVT is easy to detect on angiography when it occludes the posterior part of the entire SSS, both LSs of the deep cerebral veins, but it can be misdiagnosed with hypoplasia or aplasia when the anterior third of the SSS or the left LS are thrombosed [1,12,97]. In such situations, it is mandatory to detect other imagistic signs: thrombosis of another venous channel or delayed emptying and dilatation of collateral veins in occlusion of the anterior part of the SSS and total lack of opacification of the whole sinus or its sigmoid portion in LS thrombosis. The detection of different collateral veins usually is a hallmark of SSS occlusion and is noted in nearly half of CVT patients [1]. This technique has some limitations: it does not indicate the clot itself, it is invasive, the associated risk of surgery complications, radiation exposure, needs homogeneous teams of experts who act in dedicated hospital departments, and some subjects are allergic to the iodine contrast material [1,12,97].

##### Direct Cerebral Venography

This technique is done during endovascular therapeutic techniques, detecting the clot within the lumen either as an intraluminal filling defect (no occlusive thrombosis) or as a complete no filling (occlusive thrombosis); complete thrombosis may also indicate a “cupping appearance” within the sinus [12,97].

While the inter-observer agreement for a diagnosis of CVT is not perfect, the association of conventional contrast angiography plus brain MRI has a higher inter-observer agreement than angiography alone (94% vs. 62%) [6,97].

#### 3.6.6. Transcranial Doppler Ultrasonography (TCD) and Transcranial Color-Coded Duplex Sonography (TCCDS), or Transcranial Power Imaging, with or without the Use of a Contrast Agent

They represent the initial method used when CVT is clinically suspected in neonates and infants with opened fontanelles, identifying different aspects, depending on the size of the fontanelle: echogenic clot, intracranial hemorrhage, cerebral edema, or hydrocephalus [98].

## 4. Conclusions

CVT adult patients are younger, usually female, and present diminished frequencies of classical vascular risk factors when compared with patients with the arterial occlusive disease.

The major risk factors for CVT in adults are prothrombotic conditions, either genetic or acquired, oral contraceptives, pregnancy and the puerperium, malignancy (including myeloproliferative diseases) infection, and adenovirus-based COVID-19 vaccine-induced thrombocytopenia and thrombosis.

The pathophysiology of CVT determines the clinical spectrum and the abnormal imaging aspects. The wide clinical spectrum of CVT, often with a misleading presentation, may cause delays in diagnosis, and it is frequently represented by headache or intracranial hypertension, seizures, focal neurological deficits, and/or encephalopathy.

Both CT-CTV and MRI-MRV are excellent techniques to diagnose CVT and detect different complications. They may be combined for efficient assessment in complex or equivocal cases and must be associated with ultrasound in neonates.

## Figures and Tables

**Figure 1 life-12-00717-f001:**
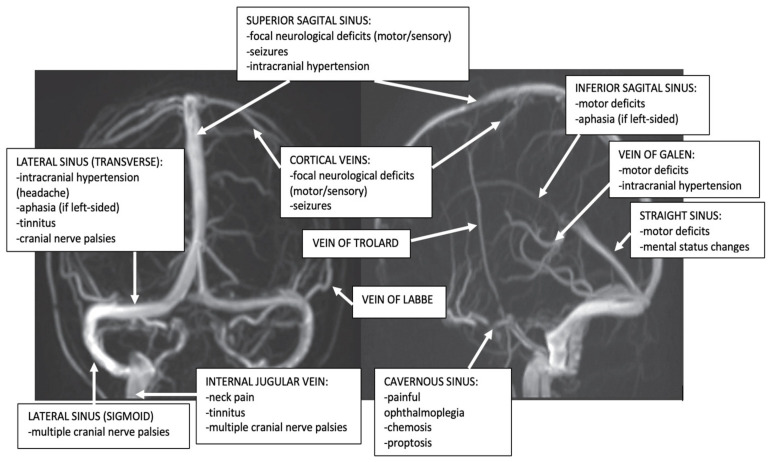
Dural sinuses and cerebral veins anatomy and major clinical syndromes according to the topography of CVT (archive of the First Department of Neurology, Clinical Emergency County Hospital, Timisoara, Romania).

**Figure 2 life-12-00717-f002:**
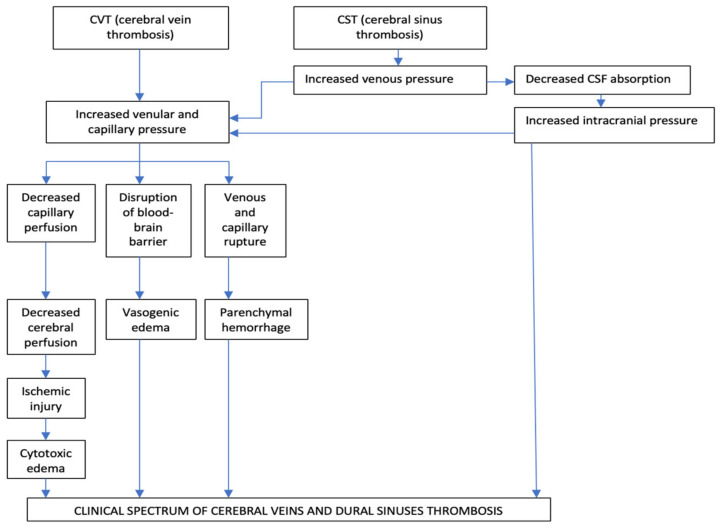
Pathophysiology of cerebral veins and dural sinuses thrombosis.

**Figure 3 life-12-00717-f003:**
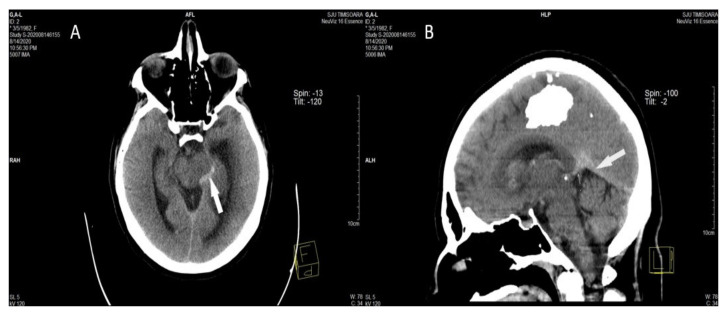
A, B. Axial (**A**) and MPR sagittal (**B**) non-contrast head computed tomography performed in the acute phase shows hyperdense appearance (acute thrombosis) of the left lateral mesencephalic [58].

**Figure 4 life-12-00717-f004:**
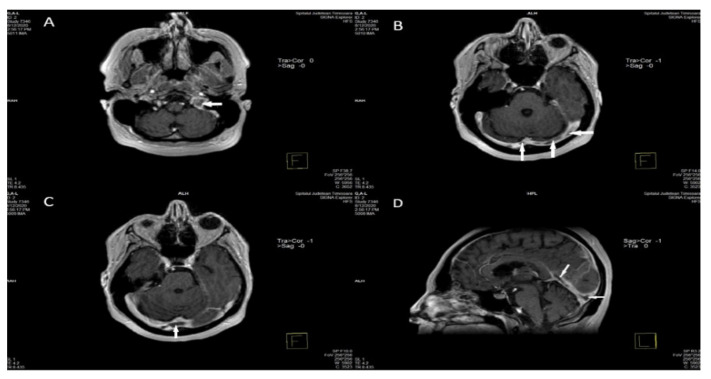
A, B, C, D. Axial and sagittal T1 post-contrast magnetic resonance demonstrate extensive filling defects throughout the dural sinuses (arrows-left sigmoid and jugular bulb (**A**), left and right transverse sinuses (**B**), sinus confluence (**C**), straight sinus (**D**)) [58].

**Figure 5 life-12-00717-f005:**
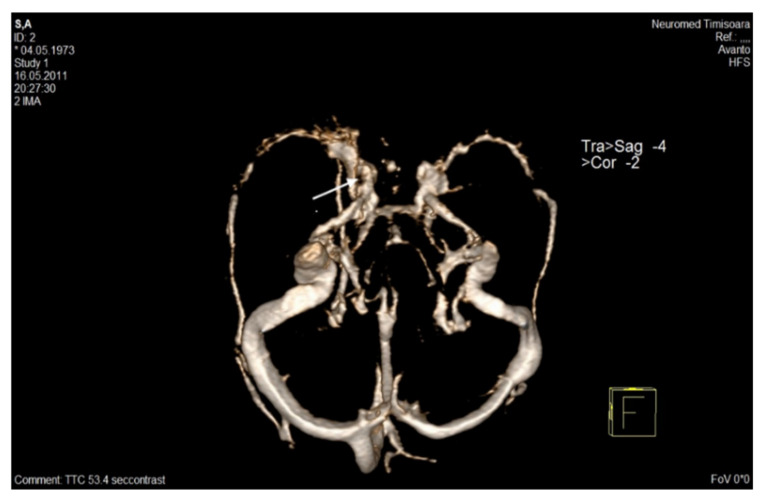
CE–MRA, venous time—VRT reconstruction in the axial plane: lacunar image in right cavernous sinus (arrow). CE–MRA: Contrast-enhanced–magnetic resonance angiography; VRT: Volume rendering technique [81].

**Table 1 life-12-00717-t001:** CT and MRI features in CVT.

Direct Signs	Indirect Signs
-Dense triangle sign (acute clot in dural sinus on NCECT) -Replacement of normal dark flow void with a clot on MRI -Cord sign (acute thrombosed cerebral vein on NCECT) -MRI equivalent: acute thrombosed cerebral vein on T2* GRE images or T2*SW images. -Empty delta sign (chronic clot in dural sinus) on CTV/ Contrast Enhancement-MRV	-Cerebral edema (on CT or MRI-T1 WI/T2WI) with elevated or mixed diffusion characteristics (on DWI) -Hemorrhagic infarction -Subarachnoid hemorrhage -Subdural hemorrhage

**Table 2 life-12-00717-t002:** Comparison of CTV and MRV in CVT.

Imaging Method	Advantages	Disadvantages
CTV	-More widely accessible than MRV -Generally costs less than MRV -Faster image acquisition than MRV -More suitable for unstable patients -Less prone to motion artifact -Better detection of cerebral small vessels	-Radiation risk -Higher rate of adverse reactions to Iodinated contrasts, including the risk of contrast-induced nephropathy -Potentially reduced visualization of skull base structures in 3D display -Acute thrombus, which is hyperdense, may mimic opacified sinus resulting in false-negative results
MRV	-No radiation risk -Low rate of adverse reactions to Gadolinium -Indicated in cases with severe renal failure if done without contrast enhancement contrast technique) -Higher sensitivity for small parenchymal lesions	-Contraindicated in cases with ferromagnetic devices and most pacemakers -More prone to motion artifact -TOF-MRV may present false-positive results from a flow that has a parallel direction with the acquisition plane. -For this reason, phase contrast MRV has to be used to identify the thrombus. -Stenotic, hypoplastic, or aplastic dural sinuses may be misdiagnosed as CVT.
CTV and MRV	Noninvasive imaging methods with indirect signs possible to detect	-Inferior resolution to detect the patency of the posterior part or entire SSS, both LSs of the deep cerebral veins to DSA

## Data Availability

First Department of Neurology, Pius Brînzeu Emergency County Hospital, Timisoara, Romania; Department of Multidetector Computed Tomography and Magnetic Resonance Imaging, Neuromed Diagnostic Imaging Centre, Timisoara, Romania.

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
