# Peer review of "An Integrated Approach on the Diagnosis of Cerebral Veins and Dural Sinuses Thrombosis (a Narrative Review)"

_life, 2022, doi:10.3390/life12050717_

Round 1
Reviewer 1 Report
This manuscript provides a detail review of the diagnosis of cerebral vein thrombosis. The authors have published on the subject before. It focuses on anatomy, etiology-risk factors, pathophysiology, clinical presentation, and diagnostics.
Overall, the manuscript is comprehensive in its discussion of anatomy, pathophysiology, clinical features and imaging diagnostics. However, they are too lengthy and often repetitive; it could be made more concise and focused. The anatomy can be greatly helped by inclusion of illustrations. Illustrations demonstrating the different between CT and MRI in the diagnosis of CVT and tables summarizing the key messages of the other sections such as pathophysiology, clinical spectrum and diagnostics would also be quite helpful.
The etiology-risk factor section is most problematic; it is amateurish and full of inaccuracies. This section can be simplified and abbreviated by deleting much of the unwarranted details concerning the diagnostics of venous thrombosis. Those are better left to hematology publications. On the other hand, quite strangely, CVT in the newly recognized syndrome of adenovirus-based COVID-19 vaccine associated thrombocytopenia and thrombosis is not mentioned.
The Conclusions section is poorly constructed. This section needs a complete revision to more informatively summarize the key take-home messages.
Bibliography: This section is quite unhelpful and often misleading; the cited articles often are not representative of the current state of knowledge, or simply fail to provide relevant, convincing data. It should be completely over-hauled.
The text also needs improvement for accuracy and conciseness.
Some examples of problematic sentences/sections are mentioned below.
Line 228 “….testing should evaluate especially for the factor V Leiden mutation, factor II gene mutation (the prothrombin variant: PT G20210A, and the homozygosity for MTHFR C677T. [5, 21-25]”
Firstly, “.. testing should evaluate especially…” is unwarranted strong. It should be properly worded to accommodate the fact that there are populations in the world where factor V Leiden and factor II G20210A alleles are non-existent except in the extremely rare cases with Caucasian ancestry. In these populations, testing for these alleles should only be targeted to those individuals. No convincing data in support of MTHFR C677T testing can be found in the cited references. (For example, #25 compares the homocysteine levels among the groups with different PAI-1 polymorphisms.) Also, protein C, protein S and antithrombin III is not included.
At a different level, the role of testing for genetic thrombophilia in clinical practice remains controversial. If the authors have any evidence indicating detection of genetic thrombophilia affects management or prognosis, it should be mentioned to justify the strong recommendation of testing genetic thrombophilia.
Finally, recommendation on diagnostics should be presented in the Diagnosis section, not in this section.
Line 240 “Another frequent mutation associated to CVT is the homozygosity for methylenetetrahydrofolate reductase gene mutation (MTHFR C677T). [2, 5, 24]”
A review of #24 shows that the study does not provide convincing evidence for “association of homozygous MTHFR C677 to CVT”. The study does not even include a proper control group for comparison. The other two citations are merely book chapters. If they include study reports with credible data, those reports should be directly cited here.
Line 245 “Acquired thrombophilia are represented in order by:”
This section incorrectly includes antithrombin III and proteins C and S. The impact of acquired low levels of antithrombin III, protein C or protein S in thrombosis risk remains uncertain. Hence, in making the diagnosis of antithrombin III, protein C, or protein S, deficiency, acquired causes are excluded. In fact, this article does not need to discuss how to make the diagnosis of protein C, protein S or antithrombin III deficiency, as the authors are obviously unfamiliar with the topic.
Line 246 “Antiphospholipid antibodies syndrome (anticardiolipin antibodies, lupus anticoagulant)”
The diagnostic term should be antiphospholipid antibody syndrome, or, more commonly, antiphospholipid syndrome, but not antiphospholipid antibodies syndrome.
The parenthesis “(anticardiolipin antibodies, lupus anticoagulant)” is inaccurate and unnecessary. It should be deleted.
“Line 248 “…. consist in anticardiolipin antibodies and antibodies against β2-glycoprotein.”
It is missing antibodies of phosphatidylserine. Again, tin this manuscript it is not necessary to get into the details of how to make the diagnosis of antiphospholipid syndrome.
Line 248 “They [antiphospholipid antibodies] were observed especially in cases with systemic lupus erythematosus (SLE), being associated with a higher arterial and venous thrombosis risk (lupus anticoagulant).”
This sentence is awkward and confusing, reflecting a lack of full knowledge of the distinction among antiphospholipid antibodies, lupus anticoagulants, and antiphospholipid syndrome. The authors should simply refrain from overstretching on to the subject of how to make the diagnosis of antiphospholipid syndrome.
Line 251 “Their deficiency produces a state of hypercoagulability and can be hereditary or determined by different acquired diseases (under 9% of cases). [4, 5] On one hand, inherited deficiency of antithrombin III is an autosomal-dominant disease, and inherited deficiencies of proteins C and S are both vitamin K dependent anticoagulants. On the other hand, acquired deficiencies of antithrombin III, proteins C and S may be associated with acute thrombosis, anticoagulation, OC, or pregnancy, and isolated acquired deficiencies of antithrombin III with liver diseases or renal loss (nephrotic syndrome). [4, 5]”
This section is confusingly incoherent and unnecessary.
For example, the sentence incorrectly gives the impression that inherited deficiencies of proteins C and S are not autosomal dominant.
Secondly, the thrombosis risk of acquired deficiency of antithrombin III, protein C or protein S remains uncertain. In practice, the diagnosis of deficiency of antithrombin III, protein C or protein S is only made after acquired causes of low levels are excluded.
Thirdly, the second part of the second sentence does not make sense. How can “inherited deficiencies of proteins C and S” be “both vitamin K dependent anticoagulants”?
Fourthly, the list of low protein C, protein S, or antithrombin III level is far from being complete. For this article, there is simply no need to mention the list of acquired causes that the authors are obviously unfamiliar with.
Line 259 “Hyperhomocysteinemia (HHcy) is an inherited or acquired disease, consisting in an augmented plasma homocysteine (Hy) level, which induces a toxic effect on the vascular endothelium affecting the clotting cascade, with an augmented risk for deep venous thrombosis (DVT), arterial occlusive disease, and CVT. [5,26,27] Inherited forms of HHcy are induced by different genetic mutations in enzymes involved in Hcy metabolism, including the mutation of the MTHFR C677T gene, which is the most frequent. Acquired forms of HHcy are produced by decreased plasma levels of folic acid, vitamin B6 or vitamin B12. [5, 26, 27]”
This paragraph is speculative, incomplete and does not correctly represent the current state of the literature.
Firstly, the role of hyperhomocysteinemia in causing venous thrombosis or arterial occlusive disease remains controversial. A review of 15 published randomized controlled trials involving 71,422 participants fails to demonstrate that using vitamin B6/B9/B12 supplementation to lower the blood homocysteine level is effective in preventing cardiovascular events. (DOI: 10.1002/14651858.CD006612.pub5)
Secondly, blood homocysteine level is affected by, in addition to MTHRF polymorphism and low levels of folic acid, vitamin B6 or vitamin B12, a variety of other factors such as age, smoking, alcohol drinking, lack of fruits/vegetables in foods, certain medications, oral contraceptives or estrogen-hormonal therapy, low physical activity, cystathionine beta-synthase mutation, and impaired kidney or thyroid function, Many of the studies on plasma or serum homocysteine levels in the literature are not properly controlled for these compounding factors, consequently producing exceedingly wide ranges of the test results. Those study results are likely to be skewed.
Line 346 “In children, the risk factors differ: a) in neonates: perinatal complications, including dehydration are the most frequent factors; b) in older children, are common especially different infections (otitis), vasculitis, hematologic disorders, and other malignancies. Thrombophilia is common in all children. [2, 5]”
This section is unintelligible.
Line 654 and on 7. Laboratory Data. Thrombophilia Testing
Classes and levels of evidence and their significance should be defined.
Line 686 “…..antiphospholipid syndrome (lupus anticoagulant, anticardiolipin antibodies)”
The parenthesis is incomplete; it should also include anti-beta 2 glycoprotein 1 and anti-phosphatidylserine antibodies). In fact, the parenthesis may be deleted. This article does not need to deal with the detail of how to make the diagnosis of antiphospholipid syndrome.
Line 687 “Other tests are represented by a functional plasminogen assay and qualitative testing of platelet functioning. [2, 5] “
Studies with data supporting these recommendations should be provided. It should be noted that plasminogen deficiency typically causes pseudomembranous lesions such as ligneous conjunctivitis and ligneous gingivitis; plasminogen deficiency has not been convincingly linked to thrombosis. It is highly unlikely that hyperactive platelet activity causes venous rather than arterial thrombosis.
Line 691 “Protein C, S, and antithrombin levels may be modified by oral contraceptives, pregnancy, severe liver disease, L-asparaginase, chemotherapy and nephrotic syndrome. [2, 4, 5]”
The statement is simplistic and incomplete. As mentioned earlier, there is no convincing evidence to indicate that low levels of protein C, protein S or antithrombin III due to acquired causes increase the risk of venous thrombosis. Again, this article does not need to deal with the complex topic of how various acquired conditions affect the levels of natural anticoagulants.
Author Response
Dear Reviewer,
Thank you for your comments and suggestions!
-Overall, the manuscript is comprehensive in its discussion of anatomy, pathophysiology, clinical features and imaging diagnostics. However, they are too lengthy and often repetitive; it could be made more concise and focused. -We tried to make the revised manuscript more concise and focused.
-The anatomy can be greatly helped by inclusion of illustrations. -We included Figure 1. Dural Sinuses and Encephalic Veins Anatomy and main clinical syndromes
-Illustrations demonstrating the different between CT and MRI in the diagnosis of CVT
-We included illustrations demonstrating the different between CT and MRI in the diagnosis of CVT in table 1 and table 2.
-tables summarizing the key messages of the other sections such as pathophysiology, clinical spectrum and diagnostics would also be quite helpful.
-we inserted the key messages concerning the Pathophysiology of CVT in figure 2.
-we inserted the key messages concerning the Clinical spectrum of CVT in figure 1.
-we inserted the key messages concerning the diagnosis of CVT in figure 6.
-The Conclusions section is poorly constructed. This section needs a complete revision to more informatively summarize the key take-home messages.
-We revised the section of Conclusions
-On the other hand, quite strangely, CVT in the newly recognized syndrome of adenovirus-based COVID-19 vaccine associated thrombocytopenia and thrombosis is not mentioned.
-We added the demanded paragraphs 4.6.COVID-vaccination-associated CVT pg 7).
-The etiology-risk factor section is most problematic; it is amateurish and full of inaccuracies. This section can be simplified and abbreviated by deleting much of the unwarranted details concerning the diagnostics of venous thrombosis. Those are better left to hematology publications.
-We revised the sections of Etiology-Risk factors and of Laboratory data.
-The text also needs improvement for accuracy and conciseness.
-we improved the text
-Bibliography: This section is quite unhelpful and often misleading; the cited articles often are not representative of the current state of knowledge, or simply fail to provide relevant, convincing data. It should be completely over-hauled.
-we modified the bibliography
Reviewer 2 Report
This paper is a complete compendium of current knowledge on cerebral veins and dural sinuses thrombosis, based on an accurate review of the literature. It discusses all aspects of the problem, not only the diagnosis and provides useful suggestions for clinical practice. It is well written and explicative. In my opinion it may be accepted for publication with minimal revision of the english language.
Author Response
Dear Reviewer,
Thank you for your comments and suggestions!
We revised the English language of our paper.

Reviewer 3 Report
The authors present an interesting article on cerebral sinus and vein thrombosis. Some aspects should be considered before accepting the manuscript for publication
-when addressing the occipital sinus, the authors should address the patency of only 10 percent in the adult population and provide adequate references.
-While the article is well written, it should be thoroughly checked for typos and wording.
-you state that 70% of the CBV are contained by the dural veins. Please provide a reference
-Please add an entire paragraph on the incidence, risk factors, epidemiology, surgical considerations and outcomes of COVID-vaccination-associated cerebral vein thrombosis.
-please include and discuss PMID 34202817
-please include and discuss PMID 15858188
-Towards the end of the manuscript, the flow while reading is somewhat reduced. Please avoid single-sentence paragraphs and revise the manuscript accordingly.
Author Response
Dear Reviewer,
Thank you for your comments and suggestions!
-when addressing the occipital sinus, the authors should address the patency of only 10 percent in the adult population and provide adequate references.
- we noted in the revised form of our paper that the occipital sinus is patent in less than 10% of adults, and we included an adequate reference. [68]
-While the article is well written, it should be thoroughly checked for typos and wording.
-We checked our article for typos and wording.
-you state that 70% of the CBV are contained by the dural veins. Please provide a reference
- The statement: “The cerebral veins contain 70% of the encephalic blood volume” belongs to Caplan LR -reference [10]
-Please add an entire paragraph on the incidence, risk factors, epidemiology, surgical considerations and outcomes of COVID-vaccination-associated cerebral vein thrombosis.
-We added the demanded paragraphs (4.6.COVID-19-vaccination-associated CVT pg 7). It contains definition, epidemiology, risk factors, pathogenesis, clinical and imagistic features. Because our review is about diagnosis of CVT, the considerations about treatment and outcomes will be included in another paper.
-please include and discuss PMID 34202817
We included and discussed PMID 34202817 (Gessler, et al) in 4.9 [76]
-please include and discuss PMID 15858188
We included and discussed PMID 34202817 (Stam) in: 3.Epidemiology, 4.Etiology-Risk Factors, 5. Pathophysiology, 6. Clinical Spectrum, 8. Imaging
-Towards the end of the manuscript, the flow while reading is somewhat reduced. Please avoid single-sentence paragraphs and revise the manuscript accordingly.
-we revised the end of the manuscript, avoiding single-sentence paragraphs.

Reviewer 4 Report
This review aims to discuss different aspects regarding cerebral venous thrombosis (CVT), including pathophysiology, clinical presentation, and neuroimaging.
In my opinion, this review may be interesting, but several modifications should be made.
Several paragraphs are too long and should be shortened. In particular, paragraphs 6 and 8. Moreover, the same paragraphs are confused: they are structured in subsections without numbers, alternating the bold with normal text. Please, follow the authors' guidelines.
Finally, in order to improve the novelty of the manuscript, it should be inserted a paragraph about CVT and COVID-19 vaccination (i.e. DOI: 10.1016/S0140-6736(21)01608-1; DOI:10.3324/haematol.2021.279075).
Author Response
Dear Reviewer,
Thank you for your comments and suggestions!
-Several paragraphs are too long and should be shortened. In particular, paragraphs 6 and 8. Moreover, the same paragraphs are confused: they are structured in subsections without numbers, alternating the bold with normal text. Please, follow the authors' guidelines.
-we reduced paragraph 6, concerning Clinical Spectrum of CVT.
We divided it in two sections 6.1.Clinical syndromes, and 6.2. Topographic diagnosis.
Both sections 6.1 and 6.2. were also divided in different subsections.
-we reduced paragraph 8, concerning Imaging of CVT.
We divided it in six sections 8.1.CT; 8.2 CTV; 8,3 MRI; 8.4 MRV; 8.5. Cerebral Intra-arterial Angiography with venous phase imaging and Direct Cerebral Venography, and 8.6. Transcranial Doppler ultrasonography.
-Finally, in order to improve the novelty of the manuscript, it should be inserted a paragraph about CVT and COVID-19 vaccination (i.e. DOI: 10.1016/S0140-6736(21)01608-1; DOI:10.3324/haematol.2021.279075).
-We inserted a paragraph (4.6.) concerning COVID-vaccination-associated CVT- pg 7.
-DOI: 10.1016/S0140-6736(21)01608-1 and DOI:10.3324/haematol.2021.279075 were used and included in references.

Round 2
Reviewer 1 Report
The manuscript remains too lengthy, with many instances of redundancy or repetition. In particular, texts related to tables and figures should be more concise.
Conclusions: Major risk factors should mention acquired prothrombotic conditions such as antiphospholipid syndrome, myeloproliferative diseases, paroxysmal nocturnal hemoglobinuria, and heparin or adenovirus based COVID-19 vaccine induced thrombocytopenia and thrombosis. The third paragraph should be more direct and concise.
Literature citations: There are too many unnecessary citations. A review article should only be cited when it reviews more than a few study publications on a controversial issue, hence obviating the need to cite all those original publications.
Figure 1 does not demonstrate all the key structures mentioned in the text. Its source should be acknowledged.
Table 1 should not be divided in rows that misleadingly imply correspondence across the columns.
Figure 5: The arrow is difficult to visualize
Language: There are many occasions of problematic language structure that are grammatically challenged and difficult to follow or comprehend. The manuscript should undergo a professional linguistic review and revision.
Author Response
Dear reviewer,
We have taken your suggestions on board and have made the following amendments:
-Figure 5: The arrow is difficult to visualize. - We modified figure 5, to include a new white arrow.
-Table 1 should not be divided in rows that misleadingly imply correspondence across the columns.
-We modified table 1, for better clarity
-Figure 1 does not demonstrate all the key structures mentioned in the text. Its source should be acknowledged.
-We modified figure 1 to acknowledge its source. It belongs to our archive (3. First Department of Neurology, “Pius Brânzeu” Clinical Emergency County Hospital, TimiÈ™oara, Romania)
-Conclusions: Major risk factors should mention acquired prothrombotic conditions such as antiphospholipid syndrome, myeloproliferative diseases, paroxysmal nocturnal hemoglobinuria, and heparin or adenovirus based COVID-19 vaccine induced thrombocytopenia and thrombosis. The third paragraph should be more direct and concise.
-We modified the second paragraph of Conclusions concerning the major risk factors for CVT according to Saposnik and to Ferro.
-We modified the third paragraph of Conclusions
-In particular, texts related to tables and figures should be more concise.
-We modified these texts to be more concise.
-The manuscript remains too lengthy, with many instances of redundancy or repetition.
-We cut down on unnecessary text.
-Language: There are many occasions of problematic language structure that are grammatically challenged and difficult to follow or comprehend. The manuscript should undergo a professional linguistic review and revision.
-The manuscript underwent a professional linguistic review and revision.
Literature citations: There are too many unnecessary citations. A review article should only be cited when it reviews more than a few study publications on a controversial issue, hence obviating the need to cite all those original publications.
-We modified the references according to Ferro JM, Canhão P. Cerebral venous thrombosis: Etiology, clinical features, and diagnosis.UptoDate: Oct 15, 2021.

Reviewer 4 Report
The authors have substantially improved their manuscript. I have only a minor suggestion:
- Please, remove figure 1 from the subheading, inserting it in the main text.
Author Response
Dear Reviewer,
Thanks again for your constructive comments and suggestions!
- Please, remove figure 1 from the subheading, inserting it in the main text.
- We removed figure 1 from the subheading, inserting it in the main text (pg. 3)
